PREREGISTERED RESEARCH ARTICLE

# Impacts of parental age and inbreeding on fitness in a wild insect

Tom Tregenza[1]*, Rolando Rodríguez-Muñoz[1], Alfredo F. Ojanguren[2], Paul Hopwood[1], Jelle J. Boonekamp[3]

**1** Centre for Ecology & Conservation, University of Exeter, Penryn Campus, Penryn, United Kingdom, **2** Departamento de Biología de Organismos y Sistemas, Universidad de Oviedo, C/ Catedrático Rodrigo Uría s/n, Oviedo, Spain, **3** School of Biodiversity, One Health & Veterinary Medicine, University of Glasgow, Glasgow, United Kingdom

* T.Tregenza@exeter.ac.uk

## Abstract

Parental age and inbreeding have both been shown to have substantial fitness effects in laboratory experiments and in observations of wild animals. These demographic effects are likely to be strongly impacted by habitat fragmentation and warming temperatures, so understanding them is a priority. In insects and other ectotherms, some processes implicated in senescence are dependent on temperature. Anticipated changes in climate may therefore have direct effects on senescence in insects, or indirect effects via parental age. Similarly, although effects of inbreeding are well studied in wild vertebrates, information about how matings between relatives affect fitness in invertebrates comes almost exclusively from laboratory studies. To bridge the divide between field studies of vertebrates and laboratory studies of insects, we conducted an experiment using wild field crickets, *Gryllus campestris*. We experimentally manipulated the relatedness of parents, their age at reproduction and the temperature they experience as they aged. We then released the offspring of these parents into a natural meadow and used a network of video cameras to monitor their adult behavior and life history throughout the course of their breeding season. We found no effect of parental age on their offspring. There were effects of inbreeding, but they were restricted to more inbred females mating to fewer males, and more inbred males being slightly smaller than outbred males. Our study suggests that effects that can be detected in laboratory studies may have relatively modest effects on fitness in nature.

## Introduction

Why organisms deteriorate as they get older and the factors that shape these senescent declines are areas of intense research [1]. One aspect that has recently come to the fore is the potential for parental age to affect physiological parameters associated with senescence in offspring [2–4]. The most widely recognized form

**Data availability statement:** All data are deposited at: https://figshare.com/s/dd51d9b2e9a824d0077e.

**Funding:** This work was supported by Natural Environment Research Council (NERC) (https://www.ukri.org/councils/nerc/): NE/V000772/1 and UKRI2653 to TT and NE/X012018/1 to JJB and by the Leverhulme Trust (https://www.leverhulme.ac.uk/research-project-grants): RPG-2024-207 to JJB. The funders had no role in study design, data collection and analysis, decision to publish, or preparation of the manuscript.

**Competing interests:** The authors have declared that no competing interests exist.

of intergenerational transfer of aging is the observation of shorter life span in the offspring of older parents, which is known as the "Lansing Effect" [5]. These effects are important because optimal life history strategies, including major evolutionary transitions (such as between being univoltine versus multivoltine or semelparous versus iteroparous), depend on the costs and benefits of reproducing at different ages. Parental age effects on offspring have the potential to alter the balance of these trade-offs. Parental age effects are frequently reported, but often the design of the study means that numerous potential proximate effects cannot be distinguished [2]. These include a range of genetic, somatic and environmental effects such as increased risk of inbreeding with age [6], accumulation of DNA damage in the germline of old parents [7], changes in parental investment [8] and telomere shortening [4].

Very few studies have examined the effect of parental age on the adult life span of wild animals. The most recent review [9] identified only 2 non-human field studies (in common terns, *(Sterna hirundo)* [10] and antler flies (*Protopiophila litigata*) [11]). A slightly earlier review [12] of field studies investigating the effect of parental age on the performance of independent offspring found only studies of birds and mammals. Many field studies have examined parental age effects on offspring traits other than longevity, most commonly pre-adult survival. Ivimey-Cook and Moorad [13] identified studies of maternal age effects on pre-adult survival in 97 species. However, there were no field studies of invertebrates amongst this cohort, despite the authors concluding that insects and rotifers are the two groups where the Lansing effect is most important. Furthermore, among field studies of vertebrates (to the best of our knowledge) all have been observational and dependent on natural rather than experimental variation in parental age. This is a substantial limitation both because age-dependent intergenerational effects can arise from alternative life-history strategies in parents, and also because parents that survive to old age are not a random sample of the population (a process known as selective disappearance) [14]. To make progress in this field, we need experimental studies in which parental age is manipulated using a design in which selective disappearance can be ruled out (i.e., there is no significant parental mortality until after the oldest parents have reproduced), or where the same parents are used at young and old ages. We also need to complement the large number of laboratory insect studies (106 identified by Ivimey-Cook and Moorad [13]) with field studies that provide insights into how significant these effects are in nature.

Studies of wild insects are particularly important in relation to parental age effects, not only because of the importance of insects in all terrestrial ecosystems, but also because many temperate insects are annual. Obligate annual species provide a unique opportunity to study senescence because selection for short generation times is absent. In species with continuous generations "fast" life histories are favored because alleles that promote short generations multiply faster in the population. This means that to compare the fitness of alternative "fast" and "slow" life-history strategies requires not just counting how many offspring individuals have but also estimating population growth rate parameters which tends to be much more difficult in nature than in the lab. In annual species we can compare the fitness consequences of different life-histories by simply using parameters that reflect reproductive success.

Studies on aging in ectotherms also create the opportunity to experimentally manipulate aging. Previous studies have relied on only comparing offspring of young and old individuals. This creates the selective disappearance issue mentioned above; old individuals are not a random sample from the population, confounding effects of age and phenotypic quality. Hence previous cross-sectional studies that do not control for variation in parental quality may underestimate the effects of parental age on offspring quality. Also, previously observed effects may be due to endogenous processes within individuals, such as failures to repair DNA after cell division, or they may be due to exogenous effects such as damage caused by interactions with the environment.

Costs of inbreeding are potentially an acute issue for populations recently reduced to a small size [15,16]. Conservation biologists have paid a lot of attention to situations where a population or entire species suffers from chronic loss of genetic diversity as a result of dwindling numbers. However, less attention has been paid to the possibility that even species that retain large population sizes may have sufficient local population structure that matings between closely related individuals are common. These effects may be exacerbated by large variances in reproductive success in insects where females can lay hundreds of eggs, and stochastic environmental factors may lead to success or failure of entire broods. Even if gene flow prevents inbreeding at the population level, impacts on population viability resulting from consanguineous matings may be significant. Furthermore, because gene flow prevents purging of deleterious recessive alleles, costs of inbreeding are expected to remain high. Many terrestrial habitats have recently become highly fragmented due to human impacts on land use, which is likely to have substantially increased local inbreeding risks. Currently, we are unable to estimate how significant such effects are, because of a shortage of data on both the prevalence of inbreeding in nature and on its effects on individual fitness. We are particularly ignorant about these issues in insects, despite them being the most numerous and arguably most ecologically important terrestrial animals.

A systematic survey by Neaves and colleagues in 2015 [17] identified 703 studies of the fitness consequences of inbreeding in natural populations. There were 231 studies of animals of which 52 were studies on insects. However, of these, only two examined the consequences of matings between related individuals in wild insects. The authors identify *"the severity of inbreeding depression in natural conditions as a major research gap"* [17]. Whitehorn *and colleagues* [18] compared the survival of artificially produced inbred and outbred bumblebee (*Bombus terrestris)* colonies, and found that inbred colonies survived for less time in the field. Armbruster *and colleagues* [19] bred *Aedes geniculatus* adults via laboratory crosses with varying numbers of generations of full sib matings. They found substantial effects of inbreeding on larval survivorship, egg batch size and female pupal weight both in the lab and in cages experiencing field conditions. Consistent with these field experiments, colonies of the ant *Formica exsecta* with higher worker homozygosity have been found to be shorter lived than more heterozygous colonies [20]. At a population level Saccheri *and colleagues's* [21] study of the butterfly *Melitaea cinxia* revealed higher extinction rates in more inbred populations.

There are large number of laboratory studies of inbreeding depression in insects [17], including studies of seed beetles, whose natural environment can be approximated in the lab [22–24]. However, in general, the extent to which laboratory measurements of inbreeding reflect costs in nature is unknown. It has frequently been argued that environmental effects are likely to exacerbate costs of inbreeding [25,26] and there is some experimental evidence of this in other groups such as snails [27] and mice [28]. There is also evidence for strong environmental dependence of inbreeding effects in laboratory studies on seed beetles [23]. However, Armbruster *and colleagues's* [19] study of *A. geniculatus* adults appears to be the only insect study comparing inbreeding depression in the lab and the field; it found that the magnitude of inbreeding depression in life history traits in the lab was very similar to that observed in the field. We are desperately short of information about how great the risk of inbreeding is in wild insects, and about its potential to depress population viability. Our own work on the field cricket *G. campestris* [29], revealed that matings between close relatives do occur in the wild, even though the small population we studied was not inbred. Very high rates of inbreeding (≥50% of matings) have been observed in wild parasitoids [30,31], and inbreeding risk is likely to be widespread even in animals with large population sizes. Experimental field studies are now needed to determine the importance of these effects in wild insect populations.

Crickets have been utilized extensively for laboratory studies of inbreeding depression, with substantial negative effects of inbreeding on fitness related traits identified in a variety of species including *G. bimaculatus* [32]; *G. firmus* [33]; *Teleogryllus oceanicus* [34]; *T. commodus* [35–38]; and *Gryllodes sigillatus [*39*]*. However, despite substantial costs of inbreeding identified in all these studies, the only attempt to examine these effects in the field, aside from our 2011 study [29], has been work on *T. commodus* [35,38]. Drayton *and colleagues's* [35] study found an effect of inbreeding on temporal properties of the calling song in the laboratory. They attempted to study this in the field using playback of songs from inbred and outbred individuals. However, in the new population used for this study [38] there was an effect of inbreeding on amount of calling, but no detectable effect of inbreeding on song structure, and unsurprisingly no difference in attractiveness in relation to structure could be detected. In *G. campestris* we have established that matings between close relatives do occur in this species [29]. Given the high costs of inbreeding identified in all cricket species that have been studied, including its closest relative (*G. bimaculatus)* [32], we could expect to see inbreeding depression in *G. campestris*. Evidence for parental age effects on offspring have recently been identified in a laboratory study of the field cricket *G. bimaculatus* [40]. Older females had longer egg incubation times and lower hatching success, and when adult, their offspring were larger but died younger. Older fathers had offspring that were less likely to survive to adulthood, and there were behavioral differences between offspring of younger and older males. Inbreeding and intergenerational transfer of aging effects have the potential to interact because the negative effects of inbreeding may be exacerbated in somatically weaker offspring from older parents. This possibility has not been directly studied, but interactions between parental and grandparental age are known [41], indicating that parental age effects may interact with other non-genetic effects.

We have studied effects of age and growth rate on telomeres in *G. campestris.* We found that although terminal telomeric repeats are present and telomere length is highly heritable, it does not deteriorate with age [42]. However, there are other potential mechanisms for parental age effects. We have a lot of evidence for senescent declines in wild crickets, where we have found changes in mortality risk with age [43] and changes in phenotype, including general activity [44], feeding behavior [45], and time spent outside the burrow [45].

Using the field system of *WildCrickets* (www.wildcrickets.org), we examined the combined effect of parental age and inbreeding treatments on adult offspring in a single factorial experiment. This design has allowed us to examine potential interactions between these effects. Comparing the mating success of males that are the result of full sib matings and outbreeding matings also provided an important test of the hypothesis that females choose mates based on genetic quality [46]. We know there are significant costs of inbreeding in crickets, so inbred males are predicted to express a phenotype functionally equivalent to a "low genetic quality" male [35]. Numerous lab studies show female choice for better quality males in field crickets [47–52]. If we cannot observe these same effects when comparing inbred and outbred males despite the large scale of our study, it suggests that the biological effect size may be very small and that measuring mate choice in wild insects may be beyond the scope of all but extremely large studies.

## Methods

This study follows a design that was pre-registered with *PLOS Biology* and the Open Science Foundation in 2023 [53]. Although there were no avoidable deviations from the execution of the pre-registered design, one component of the planned study had to be modified: the release of offspring from parents remated 20 days after their first bout of egg laying. Although we successfully remated parents after 20 days and subsequently collected eggs from females, only a very small proportion of these eggs hatched (see Results). This departure from our anticipated design is indicated on Fig 1. which is an unmodified description of the study design with the missing component indicated by a hatched area.

### Design summary

Our design aimed to rear crickets that were either inbred or outbred and simultaneously were offspring of parents with differing ages. Additionally, we aimed to compare offspring from eggs laid when their parents mated for the first time, and

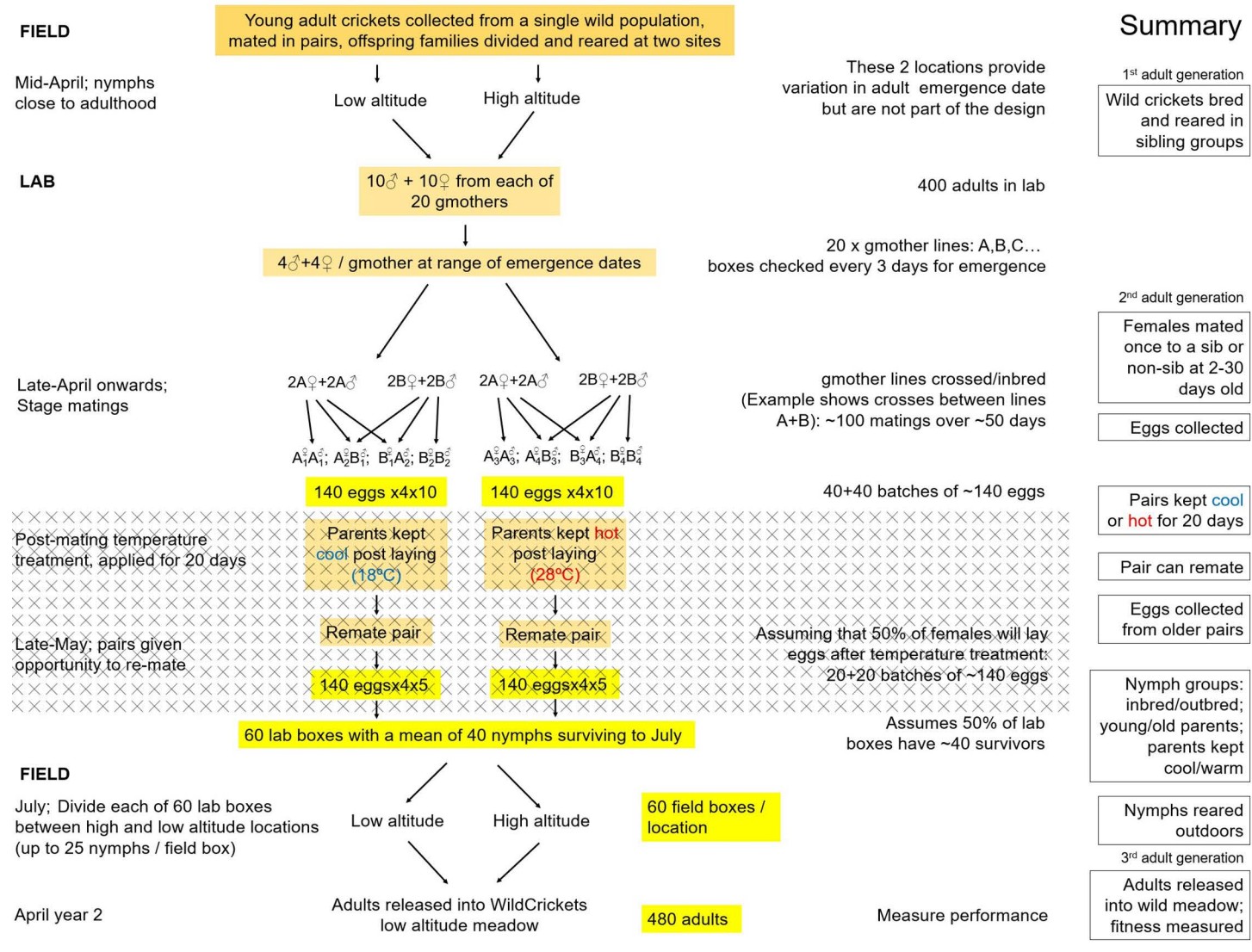

**Fig 1. Schematic of the experimental design.** The area marked with crosses indicates a component of the planned study that had to be modified (see Results).

eggs laid when their parents were at least 20 days older. In the intervening period, half of the parents were kept cool and half in warm conditions. This was designed to create a comparison between offspring of parents with the same chronological age, but where one group was expected to have experienced greater temperature-dependent physiological senescence (Fig 1). We individually tagged offspring as they became adult and released them into the wild, where we used a network of 140 CCTV cameras to record key naturally and sexually selected traits with strong links to fitness.

## Study design

We used the field cricket *G. campestris* as a model temperate insect species and measured individual fitness in nature. To produce inbred and outbred adult crickets we reared offspring that were either the result of a mating between two siblings or between two unrelated individuals from the same wild population. These parents were the offspring of young

adult females collected in the wild in 2022 and subsequently paired in 2023, once adults, with a single male. This meant that families were mainly full siblings, but with some half siblings within our sibling groups (because the females collected in the wild in 2022 might have already mated when we collected them). We collected eggs from these parents after they mated for the first time, which occurred at a range of ages from 6 to 32 days after they reached adulthood. We allowed females to lay until they had produced at least 140 eggs. At that point, the damp sand substrate in which they oviposit was removed and they were kept in isolation for 20 days. After 20 days, the egg laying substrate was returned and the female was given the opportunity to mate again with her original mate (or a sibling male in the few instances where the original male had died). In common with most insects, female crickets store sperm, so both the female and the sperm inside her were at least 20 days older at this point. During the intervening 20 days, we kept half of the parents in a large incubator at a constant temperature of 18 °C and half in another incubator at 28 °C. We adjusted lighting regimes every few days to match natural day lengths in the wild to within 30 min. Crickets were kept in identical boxes in two very similar incubators so the potential for the incubators to have a significant effect for reasons other than their temperature is extremely low. These temperature regimes meant that 20 days later, half of the individuals were expected to have experienced greater temperature-dependent physiological decline than the other half. The difference in somatic condition at the time of second mating could be confirmed retrospectively by comparing the mean remaining life-expectancy between the treatments. Once the temperature treatment had finished, crickets were maintained until their death in a well-ventilated room at ambient temperature. This approach enabled us to estimate the effects of parental age at breeding (utilizing the natural age range of producing 1st broods) and disentangle the effects of temperature dependent physiological decline (utilizing the 2nd broods produced after parental exposure to low or high temperature). We planned to use two complementary statistical modeling approaches to conduct these analyses, with the first model focusing on parental age and inbreeding in the 1st broods, and the second model focusing on the interaction between brood order and temperature treatment.

In a continuously breeding species, the effect on adult fitness of emergence date would be expected to be large. However, *G. campestris* has an obligatory winter diapause which means that in the spring, emergence to adulthood occurs over a much shorter time-period than does egg laying in the previous summer, thereby reducing or eliminating the effects of laying date on subsequent fitness. This effectively allows late-laid individuals to catch-up, so there is no *a priori* expectation that the timing of reproduction affects fitness. Nevertheless, our design controls for any effect of laying date by exploiting the fact that when females mated for the first time, this happened at a range of dates across the breeding season.

Our rearing design created inbred and outbred offspring from parents with a range of mean parental ages from 11 (when females in the lab started to lay) up to 39 days old. We kept pairs for a further 20 days and then provided them with an opportunity to remate and lay eggs after this interval. We reared nymphs in family groups, dividing each family between one of 60 field boxes at a high-altitude site and 60 at a low-altitude site. These two rearing and overwintering sites were not designed to be a specific treatment that we included to test a hypothesis; they simply represent replicates that increased the chances that sufficient individuals survive the winter and can be released into our experimental meadows the following spring. We included rearing altitude in each of the models to control for possible altitude effects. These crickets were offspring of 59 females and 54 males; we included maternal and paternal identity as a random effect in our final models. We applied treatments to parents (not grandparents) so there are no issues with pseudoreplication in our design. Offspring from the same grandparents are represented in all treatment groups so our design provides the opportunity to partition genetic variance in the traits we have measured, but that analysis was not the target of this study.

In the spring of the 2024, we released 493 (our pre-registered target was 480) adult offspring of the parents with controlled age and relatedness status into the natural habitat of our *WildCrickets* meadow. We collected detailed information on fitness related traits by tagging every individual and observing their behavior and life histories over their entire adult lives (see details below).

## Study system and specific methods for trait measurement

*G. campestris* is a flightless Western European field cricket found predominantly in dry unimproved grassland [54]. Individuals live for up to 14 months with no overlap of adult generations. Both sexes excavate burrows within a few weeks of hatching. Burrows are used as shelter from predation and bad weather but are too narrow to allow matings or other social interactions. Adult *G. campestris* are very territorial, and burrows are only shared with a single individual of the other sex during the breeding season. When outside the burrow, adults spend nearly all their time in the area just outside the burrow entrance and most interactions with conspecifics and other animals occur there. Adults feed predominantly by consuming vegetation from the immediate vicinity (within 10 cm) of the burrow mouth. Adult males call from the entrance of their burrows to attract females for mating, and both sexes move frequently among burrows during the breeding season in search of mates. New burrows may be excavated during the breeding season. Every time two adult males meet at a burrow, either one of them immediately leaves the burrow or a fight ensues until one of them takes over the burrow [29,55,56]. Females lay eggs in the ground, predominantly at or around the entrance of their burrow. Eggs hatch over the spring/summer and nymphs develop until early autumn. At their penultimate instar, nymphs enter an obligatory diapause which continues over the coldest winter months. They resume activity at the end of the winter, and in our location in northern Spain they finish their development and emerge as adults in late April/early May.

Over the last 16 years of developing our *WildCrickets* system, we have pioneered multiple new approaches to monitoring the adult lives of an entire population of wild insects. The *WildCrickets* meadow is an area of unimproved inland grassland at an altitude of 60m. The meadow is managed in a similar way every year, with the grass being mowed in mid-March and again in July/August. Between August and March, the grass is kept short with additional mowing.

In April 2024, we captured all the naturally occurring crickets in our meadow (and offspring of previous translocations) and moved them elsewhere. We then released into the meadow the offspring of the pairs reared in our boxes outdoors. The proportion of individuals from the two over-wintering locations was not the same because survival was higher in one location. These rearing locations were chosen for practical reasons and not part of the treatment groups. The number of crickets we released was 308 outbred (202 and 106 reared from high and low altitude, respectively) and 185 inbred (117 and 68 from high and low altitude).

Crickets were released into the *WildCrickets* meadow within a few days of reaching adulthood. Each released cricket was weighed (±0.01 g), photographed, and marked with an acrylic tag attached using cyanoacrylate glue to the pronotum of each adult cricket. The tag had a unique 1–2 character code with a font height of 3.1 mm, which allowed each individual to be identified on the video, and was used as a reference to measure the width of the pronotum [57].

We recorded the activity around each burrow entrance continuously using a network of ~140 infrared day/night cameras. The cameras were moved around the meadow to reconfigure the setup as new burrows were dug and old ones fell vacant. The cameras were connected to several computers provided with motion activated digital video recording software (i-Catcher: i-codesystems.co.uk) so that video was only recorded when movement was detected around the burrow. Because the number of burrows with crickets was higher than the number of cameras for about half of the duration of the experiment, we directly observed the occupants of every burrow that lacked a camera every 1–2 days by walking around the meadow and checking each burrow visually [58]. Camera movement around the meadow allowed us to capture behavioral data from most released individuals.

The adult lives of *G. campestris* involve repeated burrow swapping events [58,59], and our previous studies have shown that crickets placed in the entrance to an empty burrow immediately enter the burrow and take up residence [60]. We confined each released cricket to the immediate vicinity of its burrow for 24–48 hours by placing a small cage over the burrow entrance. When this cage was removed, the cricket had assumed full occupancy of the burrow and its behavior was completely natural. This applied to burrows previously occupied by another individual and to artificial burrows made using a metal probe. Each burrow was identified with a unique number, so that it could be read on the video. This enabled

us to identify every individual and the burrows it used and observe their adult lives in detail. We quantified key traits that have been demonstrated to relate to fitness in previous studies [61]. These traits are described in detail below.

Our previous work identified strong correlations between the values of a number of phenotypic traits and the number of offspring that individuals have in the subsequent generation [55]. We focused on four female and seven male traits:

**Body size and mass:** At the time of submission of our pre-registered report [53] our best data indicated that body size had a strong relationship with number of offspring in the subsequent generation [55,62,63], subsequent more powerful analyses using 8 years of data failed to find this relationship [61], however, it is clearly a potentially interesting variable and body size and mass have been shown to affect fitness related traits in closely related species of the same genus [48,64,65]. We weighed each individual within the first few days after emerging as an adult using a balance accurate to 0.1 mg and took a photograph that allowed us to measure thorax width [55].

**Adult life span:** Lifespan was strongly related to the number of offspring in the subsequent generation over 8 years of study of the natural population that inhabited our meadow [61]. We measured adult life span based on emergence to adulthood, and either our observation of the date of death using the video recordings (normally a predation event) or at the date when the individual was last observed alive [66].

**Number of mates/matings:** Our earlier work identified number of mates as a strong predictor of fitness in both males and females [61]. For females, we simply analyzed the number of different mates observed during their lifetime. For males we identified in advance that rather than just using the number of mates, a better predictor of offspring number is his relative number of matings per female weighted by the number of matings that female had with other males [57]). Therefore, for males, we did two analyses: First, we recorded whether or not the male mated at least once over his lifetime (mated). Second, for those males that mated, we included each unique occasion when a male and a female were observed together and recorded the number of times the male mated with the female and calculated what proportion this represented out of the total number of matings that female had with all males. This value was summed up for all the females a male mated with to create the variable proportion of matings.

Males and females often pair up at burrows and cohabit for a number of days during which time matings occur frequently [56]. Not all crickets were under camera observation all of the time, but because we moved cameras around the meadow every few days, we were able to capture a large proportion of matings as well as to identify most of the pairings. During the breeding season, we removed 28 untagged males and 52 untagged females from the meadow. These individuals are crickets that have either matured in the meadow and which we failed to find and remove before we started releasing our experimental crickets, or that moved into the meadow from surrounding grassland. Some untagged females mated to tagged males before we identified and removed them. On these occasions, for the purposes of calculating "proportion of matings" we used the median number of matings observed in tagged females (one) as an estimate of the number of matings by untagged females with other males.

**Calling effort: proportion of time calling:** Male field crickets produce a calling song by rubbing their wings together. This song serves to attract females [67,68] and is energy intensive [69]. Crickets have become a model system for studies of sexual selection because of the role of calling song in mating success. In our population, the proportion of time that males spent calling was a predictor of how many offspring they had in the next generation [55]. Calling is easy to see on the video as males raise their wings in a characteristic fashion. We estimated male calling effort by taking 1 min point samples over the 10 first minutes of every hour. If at least one of these 10 samples per hour was positive, then the cricket was recorded as calling in that hour. For each studied male, this measure provided up to 24 binary samples per day [43].

**Proportion of fights won:** When two males encounter one another at a burrow, either one of them immediately leaves the area, or there is an aggressive encounter between them. We define these interactions as fights if there is any evidence of aggression (e.g., charging, flaring of mandibles or grappling) even if the aggression is unidirectional [59]. These fights are assumed to be over potential mating partners [70], or to provide access to the safety and possible mate attraction potential of a burrow. After a fight, the loser leaves the burrow. Male dominance (his success in winning fights) was

not directly associated with higher offspring production in our earlier study [55] but did relate to his mating rate. Dominance also failed to predict mating success in a lab study of the Australian cricket *Teleogryllus commodus* [71]. However, in our meadow, fights between males are frequent and may result in injury and occasionally death [59]. Because of these evident costs, and the fact that they are easy to avoid (individuals can simply retreat from opponents) it seems likely that winning fights has the potential to bring fitness benefits. We examined all the fights that we captured on our video to determine the proportion of fights that each male won. The outcome of fights is strongly affected by the relative size of the combatants [72]. Since we also examine body size as an independent fitness variable, we include the relative size of the two males involved in each fight as a covariate in our analyses of fighting success. Fifty-two fights involved untagged individuals. In these cases, we estimated size from measurements taken from the video.

## Data analysis

As in our original design, we mated females and let them lay eggs and then transferred them to one of two temperature treatments in order test for an interaction between individual age and environmental temperature (see Fig 1). However, unexpectedly, the vast majority of the eggs from the second age period did not hatch: only two females produced offspring from this second stage of egg collection. This meant that we could not analyze within-individual differences, and instead we exploited the substantial variation we had in age among individuals using models (as set out in our design table [53]) with the general form:

$$Trait_{ij} \sim Age_j * Inbred_j + Age_j + Inbred_j + Laydate_j + Env_j + Parent_j$$

where $Trait_{ij}$ is the observed adult Trait of the *ith* observed offspring from the *jth* parent. $Age_j$ is the median age of the parental pair at egg laying. $Inbred_j$ is a factor denoting whether parents were full siblings or unrelated. $Laydate_j$ is the median egg laying date. $Env_j$ is a two-level factor denoting the offspring overwintering environment in which the cricket was reared prior to release: either high or low altitude. Although we were unable to investigate the effects of within parental age on offspring traits, our final dataset included offspring from very young (11 days post eclosion) to very old (39 days post eclosion) parents. Furthermore, there were only few parental deaths within our study period until day 39 ($n = 19$ out of $n = 196$) reducing the potential for an influence of mortality on our analyses. We are confident that our realized cross-sectional age range includes biologically old individuals as our previous studies on senescence using the same field population showed that phenotypic senescence starts as early as 12–19 days post-eclosion, and that the mean adult life span is approximately 29 days [43] showing a clear drop in survival rate after day 15 [72]. By keeping parents in captivity, we have reduced extrinsic sources of mortality, meaning our crickets live longer than in the wild, but we have no reason to believe that phenotypic senescence was delayed. Indeed, the low hatching success of second matings is a clear indication of an advanced state of reproductive senescence. Continuous variables ($Age_j$ and $Laydate_j$) were standardized to mean 0 and SD 1 to allow comparison of intercepts before running the analyses.

We ran analyses in *R*, v 4.4.3, using the package *glmmTMB*, v 1.1.10 [73], with the distribution family most appropriate to the data: thorax width and body mass data were normally distributed; other traits were non-normal or count data: for these we began by fitting Poisson distributions and tried alternative binomial family distributions testing for overdispersion using the method proposed by Harrison [74], accepting as valid, values of the dispersion parameter within the range 0.5–1.5. Following this procedure, we used negative binomial for adult life span and females' number of mates, binomial for the male traits mated and proportion of fights won, and beta-binomial for proportion of matings and calling effort.

For the analysis of mating success in males, we included male ID to account for the random effects of males mated with different females (female ID was not included since for females there is a single record for each female).

## Results

We included a total of 493 adult crickets in our analyses, 185 inbred and 308 outbred. The smaller number of inbred offspring reflects the fact that although inbred and outbred eggs and offspring were reared under identical conditions fewer offspring from inbred matings reached adulthood [75].

### Effects of parental age

Following our pre-registered analysis plan, we included parental age as the median age of both parents at the time when the first batch of eggs was laid. Because many females took several days to lay the sample of 140 eggs that we required, we calculated the laying date as the median date over all laying events until completion of the required sample. This age ranged from 11 to 39 days (22.8 ± 7.4, mean ± SD) (Fig 2). Unsurprisingly, maternal and paternal age were highly correlated (Kendall's rank correlation tau = 0.53, $z$ = 5.19). We have 3 tables describing our analyses of the effects of parental age and inbreeding on adult traits, one for female offspring (Table 1) and two for male offspring (Tables 2 and 3).

Morphology was measured as both thorax width and body mass, two traits that were highly correlated (Pearson's $r$ = 0.73, $P$ < 0.0001). These traits were not affected by parental age in either sex. The largest estimated effect size was 0.1, and variation in relation to parental age was well within random expectations (columns 1 and 2 in Tables 1 and 2). Similarly, parental age had no effect on the fitness related traits [61] of life span and number of mates in female offspring (columns 3 and 4 in Table 1); or on the male fitness traits of life span, mating (yes/no) or what proportion of his mates' total matings a male contributed (columns 3–5 in Table 2). Finally, we found no evidence that older parents had sons who sang less (column 6 in Table 2) or that were less likely to win in fights with other males (Table 3). Although as expected [72], we found that the larger of two males involved in a fight was much more likely to win (effect sizes >0.8, $P$ < 0.002) (Table 3).

### Effects of inbreeding

Inbreeding had no effect on female thorax width and body mass (columns 1 and 2 in Table 1), all effect sizes were very small (<0.03) and statistically non-significant. However, inbred males were smaller and had lower mass than outbred males (columns 1 and 2 in Table 2, values shown in Fig 3). Lifespan did not differ between inbred and outbred individuals of either sex with all effect sizes <0.11 (column 3 in Tables 1 and 2). Male mating traits and calling effort (Table 2) and

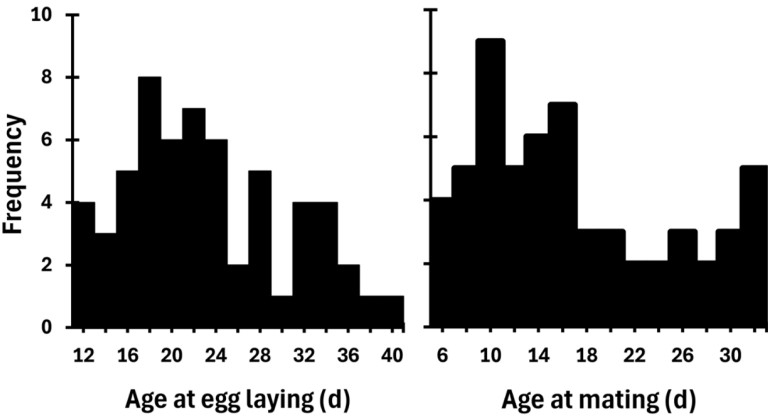

**Fig 2. Distribution of egg laying and mating ages of parents of the adults we released into our observational meadow (females, left; males, right).** Data can be found at https://figshare.com/s/dd51d9b2e9a824d0077e in GeneralDataset.txt.

**Table 1. Effects of inbreeding and parental age on female offspring: Results of GLMMs analyzing the effect of inbreeding and parental age at egg laying, on body size, life span and number of mates in _Gryllus campestris_ living in the wild. Significant P values are highlighted in bold.**

| Factor | Thorax width | | | Body mass | | | Lifespan | | | Mates | | |
|---|---|---|---|---|---|---|---|---|---|---|---|---|
| | Est. | SD | P | Est. | SD | P | Est. | SD | P | Est. | SD | P |
| Intercept | 7.337 | 0.054 | **<0.001** | 0.915 | 0.019 | **<0.001** | 2.723 | 0.054 | **<0.001** | −1.133 | 0.188 | **<0.001** |
| Inbred | −0.006 | 0.083 | 0.942 | −0.029 | 0.028 | 0.309 | −0.106 | 0.085 | 0.213 | −0.759 | 0.368 | **0.039** |
| Age | 0.032 | 0.095 | 0.735 | 0.000 | 0.032 | 0.993 | 0.017 | 0.095 | 0.855 | −0.139 | 0.336 | 0.679 |
| Laying date | −0.033 | 0.087 | 0.707 | −0.006 | 0.030 | 0.848 | −0.006 | 0.094 | 0.952 | 0.049 | 0.326 | 0.882 |
| Environment Low | 0.167 | 0.057 | **0.003** | 0.029 | 0.021 | 0.159 | −0.051 | 0.081 | 0.532 | −0.194 | 0.301 | 0.520 |
| Inbred:Age | −0.098 | 0.084 | 0.246 | −0.030 | 0.029 | 0.308 | 0.056 | 0.090 | 0.537 | 0.220 | 0.365 | 0.547 |
| | _N_ | | | _N_ | | | _N_ | | | _N_ | | |
| Observations | 272 | | | 279 | | | 279 | | | 279 | | |
| Dams | 51 | | | 52 | | | 52 | | | 52 | | |
| Sires | 46 | | | 47 | | | 47 | | | 47 | | |
| **Random effects** | **Var** | | | **Var** | | | **Var** | | | **Var** | | |
| Dams | 0.004 | | | 0.003 | | | 0.000 | | | 0.000 | | |
| Sires | 0.000 | | | 0.000 | | | 0.000 | | | 0.000 | | |
| Residual | 0.148 | | | 0.022 | | | | | | | | |

dominance (Table 3) were also similar between inbred and outbred individuals. However, inbred females tended to mate with fewer males than outbred females (final column of Table 1, Fig 4).

We found no evidence in any of the traits we studied for any interactions between whether adult crickets were inbred or not and the age of their parents (Tables 1 and 2).

## Discussion

### Effects of parental age

Laboratory studies have identified insects as particularly likely to experience inter-generational effects of parental age [13]. However, there have been few studies specifically designed for the estimation of parental age effects on offspring adult life span, and there is only one other study on a wild insect population [11]. As the field develops there is a danger that publication biases in favor of positive findings will distort our understanding, which was one of the reasons we chose to pre-register this study. We designed our study to test if parental age has negative effects on offspring adult life span, body condition, and reproductive performance, as predicted by theory [2]. We did not find any significant effects of parental age on key offspring adult behavioral and life-history traits including adult life span.

We were (unexpectedly) unable to analyze within-individual variation in parental age. This was because of almost complete hatching failure in the eggs laid by females that had already mated and laid eggs 20 days previously. We did not anticipate this issue because we have observed females in the wild laying eggs throughout their lives, and we assumed that eggs laid in later life would be viable. We can only speculate about why females that are still alive and active appear to be functionally impaired. Post-reproductive life span is often seen in zoo animals [76,77], but has received very little attention in animals where parental care is absent. It has been suggested that post-reproductive life span might be an insurance against variation in life span [78]. Our wild crickets frequently experience periods of bad weather during the breeding season which severely restricts their activity. It is possible that this capacity to survive and even continue to lay eggs is a side-effect of an adaptive capacity to pause reproduction in case of unfavorable conditions.

Despite the lack of longitudinal data on parental age, the variation in the ages at which pairs reproduced at their first attempt was considerable. Female ages at egg laying ranged across 28 days, and male ages at mating across 26

**Table 2. Effects of inbreeding and parental age on male offspring: Results of GLMMs analyzing the effect of inbreeding and parental age on body size, life span, probability of mating, proportion of matings for each mate relative to other males and calling effort in *Gryllus campestris* living in the wild. Significant P values are highlighted in bold.**

| Factor | Thorax width | | | Body mass | | | Lifespan | | | Mated | | | Proportion of matings | | | Calling effort | | |
|---|---|---|---|---|---|---|---|---|---|---|---|---|---|---|---|---|---|---|
| | Est. | SD | P | Est. | SD | P | Est. | SD | P | Est. | SD | P | Est. | SD | P | Est. | SD | P |
| Intercept | 7.582 | 0.060 | **< 0.001** | 0.899 | 0.019 | **< 0.001** | 2.719 | 0.077 | **< 0.001** | −1.280 | 0.287 | **< 0.001** | 0.362 | 0.434 | 0.404 | −2.461 | 0.167 | **< 0.001** |
| Inbred | −0.191 | 0.086 | **0.027** | −0.060 | 0.027 | **0.024** | −0.087 | 0.107 | 0.416 | 0.411 | 0.397 | 0.301 | 0.718 | 0.704 | 0.308 | 0.140 | 0.224 | 0.531 |
| Age | 0.016 | 0.097 | 0.870 | −0.002 | 0.030 | 0.951 | −0.030 | 0.124 | 0.807 | −0.406 | 0.487 | 0.405 | −0.472 | 0.733 | 0.519 | −0.132 | 0.264 | 0.618 |
| Laying date | −0.015 | 0.091 | 0.870 | −0.012 | 0.028 | 0.662 | −0.007 | 0.120 | 0.954 | 0.303 | 0.450 | 0.501 | 0.826 | 0.734 | 0.260 | 0.261 | 0.255 | 0.305 |
| Environment Low | 0.113 | 0.066 | 0.085 | 0.040 | 0.020 | **0.043** | −0.060 | 0.098 | 0.542 | −1.009 | 0.417 | **0.016** | −0.628 | 0.666 | 0.346 | 0.053 | 0.210 | 0.801 |
| Inbred:Age | −0.124 | 0.082 | 0.128 | −0.006 | 0.025 | 0.825 | 0.087 | 0.098 | 0.372 | 0.340 | 0.366 | 0.353 | −0.397 | 0.612 | 0.516 | −0.107 | 0.194 | 0.582 |
| | *N* | | | *N* | | | *N* | | | *N* | | | *N* | | | *N* | | |
| Observations | 211 | | | 214 | | | 214 | | | 214 | | | 70 | | | 202 | | |
| Dams | 51 | | | 53 | | | 53 | | | 53 | | | 28 | | | 53 | | |
| Sires | 47 | | | 49 | | | 49 | | | 49 | | | 44 | | | 49 | | |
| **Random effects** | Var | | | Var | | | Var | | | Var | | | Var | | | Var | | |
| ID | | | | | | | | | | | | | 1.210 | | | | | |
| Dams | 0.033 | | | 0.003 | | | 0.000 | | | 0.000 | | | 0.000 | | | 0.000 | | |
| Sires | 0.000 | | | 0.000 | | | 0.000 | | | 0.079 | | | 0.000 | | | 0.000 | | |
| Residual | 0.170 | | | 0.016 | | | | | | | | | | | | | | |

**Table 3. Results of a GLMMs analyzing the effect of inbreeding and parental age on dominance of their male offspring in naturally occurring fights in *Gryllus campestris* living in the wild. Significant P values are highlighted in bold.**

| Factor | Est. | SD | *P* |
|---|---|---|---|
| Intercept | −0.214 | 0.428 | 0.618 |
| Inbred | 0.253 | 0.775 | 0.744 |
| Age | −0.350 | 0.855 | 0.683 |
| Laying date | 0.642 | 0.819 | 0.433 |
| Environment Low | 0.209 | 0.685 | 0.760 |
| Relative size | 0.834 | 0.269 | **0.002** |
| Inbred:Age | −0.205 | 0.695 | 0.768 |
| | *N* | | |
| Observations | 181 | | |
| ID | 51 | | |
| Dams | 30 | | |
| Males | 28 | | |
| **Random effects** | **Var** | | |
| ID | 1.020 | | |
| Dam | 0.000 | | |
| Male | 0.762 | | |

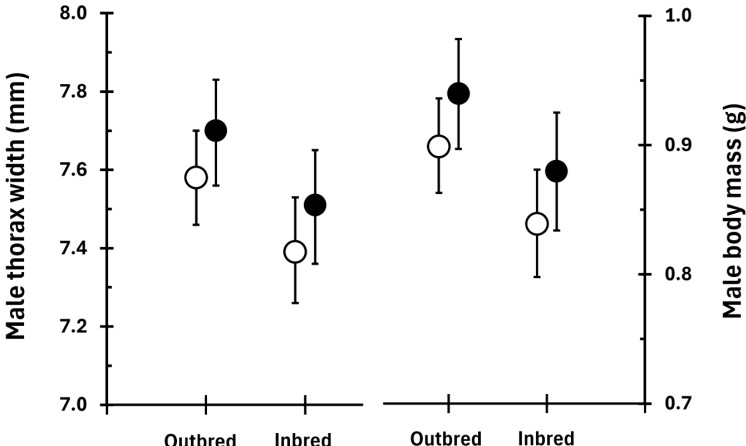

**Fig 3. The difference in body mass and thorax width between outbred and inbred males when reared at high altitude (empty circles) or low altitude (filled circles).** Means ± 95% CI obtained with the R package *emmeans* from the output of the model shown in Table 2. Data can be found at https://figshare.com/s/dd51d9b2e9a824d0077e in GeneralDataset.txt.

days (Fig 2), this is a very substantial range considering that across 10 years the average adult life span in our wild population was 29 days [43]. Treating male and female age as separate predictors rather than using mean age did not change our conclusion that parental age does not affect any of the measured traits (Data are provided for any reader wishing to confirm this conclusion). When analyzing effects of age on traits there is the potential for such effects to be masked by selective disappearance [14] leading to false negatives. In relation to our observation of a lack of any effect of parental age on offspring traits, one potential explanation could be that such effects are hidden because only higher fitness individuals survived to mate and lay eggs at an old age (i.e., the negative effects of parental age could

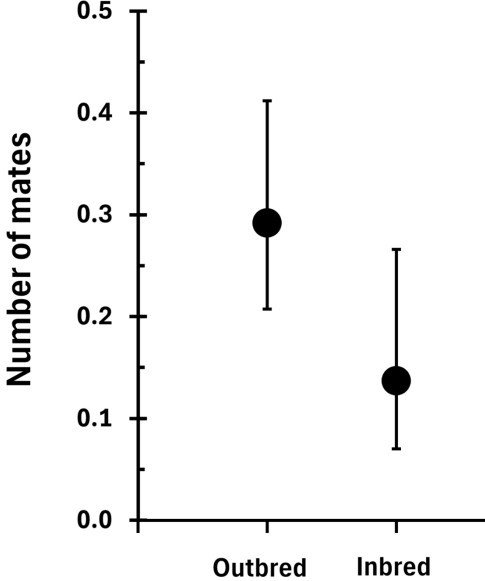

**Fig 4. Lifetime number of mates per female in relation to inbreeding status.** Means ± 95% CI obtained with the R package *emmeans*, from the output of the model shown in Table 1. Data can be found at https://figshare.com/s/dd51d9b2e9a824d0077e in GeneralDataset.txt.

be counter-balanced by older parents being of higher fitness). However, only 19 out of 196 of the individuals that were potential parents of our released crickets died before the age at mating or laying eggs of the oldest parents in our study. This low level of mortality would only be sufficient to obscure a weak true effect of parental age. Hence, we can rule out selective disappearance as an explanation for our finding that there were no negative effects of parental age in this study.

A related issue is that exploiting among individual variation in egg laying date creates the possibility that females that lay eggs later in life differ from those laying early in life. We cannot exclude this possibility but given the lack of effects of maternal age on offspring that we found, this is conservative in relation to our findings. It is also possible that there were effects of parental age on early development and overwintering survival of nymphs, but this was outside the scope of our present study. Our conclusion is that this is a study with substantial power to detect parental age effects on adults. The fact that we don't find any suggests that effects on adult offspring of parental age in this species are either small and unlikely to be evolutionarily very significant or are absent altogether

## Effects of inbreeding

We found modest evidence for effects on adults of high within-individual inbreeding coefficients in two classes of traits: body size/mass and degree of polyandry. Inbred male offspring were significantly smaller in terms of both their thorax width and their body mass at adulthood than outbred males (Fig 3) while there were no effects of the inbreeding treatment on females (Table 1). Although significant, the size of the effect of inbreeding in males was small; inbred males were 2%–3% smaller (or lighter) than outbred males (Fig 3). This either represents a sex-specific effect of inbreeding [79], or a type-I statistical error. Our analysis suggests the probability that this difference was due to chance is between 2% and 3% (Table 3). However, if we correct for investigating multiple traits, this figure will rise substantially. In our earlier study of selection on adult traits across eight independent generations of a wild population of *G. campestris* [61] we found strong directional selection on a range of adult traits; however, thorax width was not under directional selection in any of

the years that we studied. This suggests that the difference in body size we observed between inbred and outbred males might not have any effect on their fitness in the wild.

The other effect of inbreeding that we observed was a lower degree of polyandry in inbred females (Fig 4). There was modest evidence for this effect ($P=0.039$) when female traits were modeled in relation to inbreeding and maternal age (Table 1), but when inbreeding was modeled with paternal age, the support for a true effect of polyandry fell slightly ($P=0.066$). As with male body size, there is clearly a danger of a type-I error here. However, the effect is quite large (Fig 4) with inbred females having 50% fewer mates. Polyandry is associated with higher lifetime reproductive success in this species [61], suggesting a role for post-copulatory female choice as seen in *G. bimaculatus* [32,80]. Members of both sexes are frequently observed leaving a burrow that they have been sharing with a mating partner indicating that both male and female mate choice occurs. It is possible that more inbred females are less attractive to males, or that they are less likely to encounter males through moving around the meadow searching for mates. We know from previous studies [61] that males tend to have more offspring when they mate with more females, but we did not see any difference in number of mates or mating rate in males according to their inbreeding status.

As discussed above, the most recent systematic review of studies of the impact of inbreeding in animals [17] identified only 4 studies of inbreeding in insects in wild or semi-wild conditions. Subsequently, a 2025 review [81] focusing specifically on insects, highlights the fact that almost our entire understanding of costs of inbreeding in insects relates to population-level overdominance effects and losses of genetic diversity. That review [81] did not identify any more recent attempts to experimentally manipulate inbreeding in wild insects, although laboratory studies reporting costs of inbreeding continue to accumulate. For instance, although monarch butterflies (*Danaus plexippus*) do not discriminate against relatives in mate choice [82], offspring of matings between full siblings experience higher developmental mortality and shorter lifespans [83]. The authors argue that the lack of inbreeding avoidance reflects a very low probability of encounters between relatives in this highly mobile species. However, in more sedentary insects, inbreeding risk is likely to be present. We have previously demonstrated that in *G. campestris*, siblings do encounter and mate with one another in the wild [29].

Our experiment builds on two previous studies which identified fitness costs of consanguineous matings in wild bumblebees (*Bombus terrestris*) [18] and wild mosquitoes (*Aedes geniculatus*) [19]. In contrast to those studies, we found only limited evidence for lower fitness of adults in nature as a result of full-sib matings in *G. campestris*. This finding was unexpected, mainly because of evidence of negative effects on fitness traits of inbreeding in laboratory studies of several closely related cricket species [32–39]. Also, because in its closest relative *Gryllus bimaculatus* (hybrids between the two species are fertile [84]), polyandrous females are able to avoid fertilizing their eggs with sperm from closely related males [32,80], which is most easily explained as an adaptation to reduce inbreeding.

It is possible that the very large population of *G. campestris* from which we sourced the parental individuals for our study simply does not experience inbreeding depression in any of the traits that we examined. This has been observed in some animals, for instance, in Eastern mosquitofish (*Gambusia holbrooki*) [85]. However, the fact that there does appear to be substantial inbreeding depression in the pre-adult phases of our study (below) makes it more puzzling that we do not observe this in adults in natural conditions.

A potential explanation for the apparent contradiction between previous laboratory studies on crickets and our field study, is that our study focused specifically on fitness consequences in adults. In common with most insects, *G. campestris* females can lay many hundreds of eggs. We collected 140 eggs from most females and then after a period of rearing in the lab we transferred a maximum of 25 nymphs into boxes outdoors as described above. Analyzing pre-adult effects of inbreeding and parental age was not part of our pre-registered plan and will be the subject of another study [75]. However, it is important to note that of the thousands of eggs laid, only a small proportion survived to adulthood. The 308 outbred and 185 inbred lab-reared adults that were released represent a 40% reduction in the survival rate of inbred individuals relative to outbred individuals. This survival differential is likely the result of higher mortality of inbred individuals across pre-adult life history stages. It potentially results from lower viability of inbred eggs, as has been observed in

*G. bimaculatus* [32,80] where females mating to full sibling males had a 36% reduced egg hatching rate relative to out-breeding individuals [32]. Similarly, inbred *Teleogryllus commodus* eggs had a 12% reduction in hatching rate and inbred nymphs had 33% lower survival than outbred nymphs [35]. This mortality means that is it likely that there is substantial pre-adult viability selection removing poor quality individuals suffering from deleterious effects of inbreeding. However, previous laboratory studies of the costs of inbreeding in crickets suggest that effects of inbreeding are not confined to pre-adult stages. For instance 13% and 32% reductions in male and female (respectively) adult life span as a result of full sib matings have been observed in *T. commodus* [35]; a 14% reduction in female fecundity in *Gryllodes sigillatus* [33]; a 30% reduction in calling effort in *T. commodus* [38]; and a 27% reduction in competitive fertilization success in *T. oceanicus* [34]. It is therefore surprising that we did not detect any effects of inbreeding other than thorax width and the degree of polyandry as discussed above, despite substantial statistical power and reliable data on life span, mating success, and other fitness-related traits.

All our estimates of the difference between inbred and outbred individuals that are anywhere near being statistically supported (e.g., $P < 0.2$) are in the expected direction, with lower values for inbred individuals. Given our substantial sample size, which was larger than any of the previous lab studies (other than the calling effort measures in [38]), we can confidently conclude that if these inbreeding effects are true, their effect sizes are small, weaker than effects that have been detected in the laboratory. There are numerous environmental effects in nature (weather, attacks by predators, spatial variation in food resources and so-on) that might obscure differences that can be measured in the laboratory. Nevertheless, our study runs against the orthodoxy that costs of inbreeding are greater under stressful conditions [86,87]. A recent meta-analysis, (albeit focusing on sex differences in inbreeding depression [79]) also failed to find any difference in the costs of inbreeding in stressful and non-stressful environments. Our study indicates that we should not assume that if laboratory studies reveal costs of inbreeding, these effects will be present or indeed greater in the wild.

Our study highlights the general need for additional experimental field studies on intergenerational effects. The discrepancy between the findings from our field study and those of previous laboratory studies on similar organisms raises questions about the relevance of laboratory findings for wild populations. For example, it is likely that a plethora of natural hazards and other variables will interact in unpredictable ways with the effects of inbreeding and parental age. Given the growing number of field studies on senescence in wild insects [41] there is now an exciting opportunity to exploit the availability of new field systems to study intergenerational effects of senescence and inbreeding in the wild. Such studies would be uniquely capable of estimating the effect sizes of parental age and inbreeding in the natural environment. They could also determine the extent to which effect sizes in wild populations are shaped by individual variation due to stochastic environmental effects, versus a true difference in biological effects across laboratory and natural environments.

## Ethics statement

This study has been approved by the University of Exeter's Research Ethics Panel approval number 513752. All crickets used in the study either died of natural causes during rearing in the laboratory or lived out their natural lives in the wild as part of our study or were returned to the wild elsewhere.

## Acknowledgments

The authors would like to thank Emma Álvarez Alba and her family, Belarmina López, Ruonan Li, Mark Pitt, and Emily Gilford for extensive help in the field. Also undergraduate students from the University of Oviedo for help rearing crickets in the laboratory and/or assisting in the field: Mafalda Álvarez, Iván Espeso, Rodrigo Fernández, Paula Ferrer, Ana Deva García, Eva García, Fernando Ramón García, Lucía Girón, Diego González, Jimena Gonzalvo, Ainhoa Hernández, Christian Herresánchez, Víctor Llorente, Aitor López, Miguel Margolles, Gonzalo Martínez, Lucía Martínez, Pelayo Menéndez, Juan Pablo Menes, Alejandro Mieres, Marina Orduña, Daniel Peón, Pablo Pereda, Claudia Povedano, Emma Shorter y Alba Suárez.

## Author contributions

**Conceptualization:** Tom Tregenza, Rolando Rodríguez-Muñoz, Alfredo F. Ojanguren, Paul Hopwood, Jelle J. Boonekamp.

**Data curation:** Tom Tregenza, Rolando Rodríguez-Muñoz.

**Formal analysis:** Tom Tregenza, Rolando Rodríguez-Muñoz, Jelle J. Boonekamp.

**Funding acquisition:** Tom Tregenza.

**Investigation:** Tom Tregenza, Rolando Rodríguez-Muñoz, Alfredo F. Ojanguren, Jelle J. Boonekamp.

**Methodology:** Tom Tregenza, Rolando Rodríguez-Muñoz, Alfredo F. Ojanguren, Paul Hopwood, Jelle J. Boonekamp.

**Project administration:** Tom Tregenza.

**Resources:** Tom Tregenza.

**Supervision:** Tom Tregenza, Alfredo F. Ojanguren.

**Writing – original draft:** Tom Tregenza, Rolando Rodríguez-Muñoz, Jelle J. Boonekamp.

**Writing – review & editing:** Tom Tregenza, Rolando Rodríguez-Muñoz, Alfredo F. Ojanguren, Paul Hopwood, Jelle J. Boonekamp.

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
