## [Editor Report · Decision Letter 0]

27 Mar 2023

Dear Tom, 

Thank you for submitting the revised version of your manuscript entitled "Impacts of parental age and inbreeding on fitness in a wild insect" for consideration as a Preregistered Research Article by PLOS Biology.

Your manuscript has now been evaluated by the PLOS Biology editorial staff as well as by an academic editor with relevant expertise and I am writing to let you know that we would like to send your submission out for external peer review.

Once your full submission is complete, your paper will undergo a series of checks in preparation for peer review. After your manuscript has passed the checks it will be sent out for review. To provide the metadata for your submission, please Login to Editorial Manager (https://www.editorialmanager.com/pbiology) within two working days, i.e. by Mar 29 2023 11:59PM.

Kind regards,

Ines

--

Ines Alvarez-Garcia, PhD

Senior Editor

PLOS Biology

---

## [Decision Letter · Decision Letter 1]

5 Jun 2023

Dear Dr Tregenza,

Thank you for your patience while your manuscript entitled "Impacts of parental age and inbreeding on fitness in a wild insect" went through peer-review at PLOS Biology as a Preregistered Research Article. Please also accept my apologies for the delay in providing you with our decision. Your manuscript has now been evaluated by the PLOS Biology editors, an Academic Editor with relevant expertise, and by two independent reviewers.

The reviews are attached below. As you will see, both reviewers think the proposed experiments address a very interesting and timely question, but they also have several concerns that would need to be addressed. Reviewer 1 raises two main points: one related to the assumption that temperature treatment will affect the offspring only via effects on maternal aging, as it is also possible that this factor can have parental effects even in young crickets, and another to clarify if the temperature and age at breeding treatments will be applied to females only, to both sexes together or to each sex independently, as this is important to check if the effects observed will be maternal or paternal or due to the interaction of both. Reviewer 2 would like you to justify that 20-day-old crickets are indeed old and has some doubts that ‘biological age’ is a useful concept.

In light of the reviews, which you will find at the end of this email, we are pleased to offer you the opportunity to address the comments from the reviewers in a revision that we anticipate should not take you very long. We will then assess your revised manuscript and your response to the reviewers' comments with our Academic Editors aiming to avoid further rounds of peer-review.

**IMPORTANT - SUBMITTING YOUR REVISION**

3. Resubmission Checklist

Sincerely,

Ines

--

Ines Alvarez-Garcia, PhD

Senior Editor

PLOS Biology

Reviewers' comments

Rev. 1:

The proposed experiment addresses a very interesting and timely question: how parental age at breeding affects offspring phenotype and performance, and whether such effects are modulated by the ambient environment that parents experience. The effects of senescence on offspring remain poorly understood, and represent a fascinating example of nongenetic inheritance. This study is especially valuable in that it aims to investigate effects of parental age at breeding in a natural population. While a few studies have investigated such effects in the wild, very few have manipulated parental age at breeding and none have also manipulated the parental environment.

Nonetheless, I have a few concerns about the design of the proposed study.

Major comments:

I have a couple of concerns in relation to the proposed experiment design.

First, I think it's problematic to assume that the temperature treatment will affect the offspring only via effects on maternal ageing. The authors refer to individuals that are chronologically old but kept at the lower temperature as "biologically young." This is surely an assumption. It also looks like the design is oddly unbalanced: there are no O-Y-A or I-Y-A treatment combinations (Table 1), presumably because the authors don't see any reason to investigate effects of warm temperature on young-breeding crickets. But it's possible that ambient temperature would induce some kind of parental effect even in young crickets. It seems important to me to treat temperature as a factor independent of assumed effects on ageing, and include O/I - Y - A treatment combinations in the experiment. In other words, I would expose even the young-breeding crickets to both cool and warm temperature treatments, to see whether a brief exposure when young would still affect offspring phenotype and performance.

Second, the experiment description is also unclear on a fundamental point: Will the temperature and age-at-breeding treatments be applied to females only, to both sexes together, or to each sex independently? How this is set up will determine whether the results can reveal whether effects are maternal or paternal, or depend on an interaction of maternal and paternal treatment. Ideally, females and males should be exposed to treatments independently, and then crossed in all combinations (full factorial design). That would allow the authors to determine effects of maternal treatments, paternal treatments, and maternal x paternal interactions. If this is not possible for practical reasons then the next best option would be to expose females to treatments and then pair all females with males of standard age and temperature treatment. That would allow the authors to test for maternal effects per se. Exposing both sexes together will make it impossible to determine whether effects are maternal or paternal or depend on an interaction of the two.

I'm also not sure how to interpret the statements on interaction effects in this study (e.g. P9 L20 - P10 L2; P11 L18-22). The authors appear to be saying, on a priori grounds, that they do not have enough power to reliably detect interactions. Does this mean that that they will not test for interactions in their data? Or that they will test for interactions, but then disregard interactions as unreliable even if some interactions are found to have statistical support? If the authors feel that they are unlikely to have sufficient statistical power to test for interactions then why use a factorial design? If interactions are likely to be present (even if undetectable statistically), then how can this study provide a reliable test of the main effects?

Other comments:

P4 L24 - P5 L2: There are words missing. These sentences are unclear.

P12 L6-8: How were the temperatures chosen? Are they ecologically relevant?

P34 L7-9: There are indeed very few such studies, but there are some. For example, a study on rotifers investigated maternal age effects when mothers were maintained on normal vs. restricted diets (Gribble et al. 2014 Maternal caloric restriction partially rescues the deleterious effects of advanced maternal age on offspring Aging Cell 13, 623-630 ). There have also been a couple of studies on mice with manipulations of maternal diet and age at breeding. Perhaps the authors are referring specifically to the effects of temperature?

Rev. 2:

There is a massive gap in our understanding of ageing in natural populations of invertebrates, and 'Impacts of parental age and inbreeding on fitness in a wild insect' proposes a much-needed investigation into this area. The manuscript is very well written: it is straightforward, complete, and interesting, and I found no substantive omissions or inaccuracies. The hypotheses are well-supported, and the experimental design is appropriate. I do have concerns that the 20 day-old crickets are truly "old" (see comments below). This choice needs to be justified because if they are not truly old, then the study may not actually be assessing a range of ages relevant to the phenomenon of senescence. Even if this choice of 20 days cannot be justified, a greater age might simply be substituted (although there may be feasibility concerns that I am not aware of).

As far as I can tell, the design is sufficiently powered (with the caveat above). The controls appear appropriate and well-considered.

I think that this could be a valuable contribution to the ecological study of ageing, and I would very much like to see this work carried out and published.

Please see specific comments below.

Other comments:

1. p2 (6) "time" is unclear. Is this 'age' or 'date'?

2. p2 (6-7): "so changes…are poorly understood". I do not follow this logic.

3. p2 (10): "divide between field … and laboratory studies". This divide is not introduced prior to this.

4. p3 (10): may be profoundly…

5. p4 (10-11): Isn't the referenced paper [13] about juvenile survival rather than the Lansing Effect?

6. p4 (15-17): I can see how experimental manipulation could be helpful (for example, by generating/removing correlations between parental ages to improve statistical power to detect interaction effects), but I do not understand how this can ameliorate or better estimate the effects of selective disappearance.

7. p4 (24) to p5 (2): I do not follow this argument. Perhaps if the penultimate sentence (24-25 appears to be a fragment) was amended, it might become clearer.

8. p5 (11-13): I have to admit that I am a bit sceptical that 'biological age' is a useful concept. Ignoring that, I cannot see how altering condition necessarily 'resolves' complications arising from selective disappearance. Within some environment, we can expect that observed rates of aging are the summed effects of longitudinal aging and selective disappearance. If we change the environment, we might change that balance, but I should think that the most likely outcome would be to change the magnitude of both factors in the same way (i.e., a more stressful environment would both accelerate longitudinal senescence and intensify the effects of selective disappearance). Working this out could be interesting, but I see this as manipulating the environment and assessing changes in ageing, rather than directly manipulating ageing. Also, is selective disappearance a problem or a phenomenon with studying in its own right? I would lean towards the latter given that we have statistical methods that can help us measure its effects on many traits.

9. p6 (24): remove comma

10. p7 (8-10) : also see Yun and Agrawal (2014) for an interesting take on this...

11. p7 (12): "lab-as-field studies" - I do not know this term

12. p7 (end): The amount of ID experienced by a population follows from both the prevalence and degree of inbreeding and the effects of inbreeding when inbred. The potential for ID follows just from the latter. If it is true that you are assessing the latter, could you make this clear here?

13. p8 (15): was this effect an increase or decrease?

14. p9 (4-6): It is not clear to me why this necessarily leads to a weak design. Please explain.

15. p9 (9-10) : Can we say with confidence that telomere attrition has a causal effect on offspring performance? If not, perhaps noting that the relationship is correlational would be safer.

16. p9 (19-20): is this directly relevant?

17. p10 (9-11) : I do not follow this logic.

18. p11 (12-22): I think that this section of the paragraph essentially repeats elements from the end of p.9. The text could be streamlined a little to save some length.

19. p12 (5): Can you justify representing 20 day-old crickets as "old"? I appreciate that this can be a bit subjective, but I think that this is important. One way to do this might be to refer to earlier work that finds senescent declines at this age. Another might be to present evidence that cumulative survival rates have appreciably declined by the age and suggest that this implies a substantial loss of selection to maintain performance. The latter is a gambit used by Ivimey Cook and Moorads' study of maternal age effects in N. vespilloides (AmNat 2018).

20. p12 (10): This would be a daunting standard to be held to!

21. p12 (17): remove comma

22. p12 (17-21): I take this to mean that because this species does not have overlapping generations, there is no a priori expectation that the timing of reproduction affects fitness. Is this correct?

23. p20 (Table 2): Incidentally, if these are aligned in the direction of higher fitness, than they correspond to directional selection gradients.

24. p20 (3): Given that you have established that generations are non-overlapping, you can justify calling this 'fitness'. I think that this is worth pointing out since few wild animal studies can make this claim.

25. p21 (4-6): Does this average take into the account of censoring? Also, how does your assessment of 20-day-olds as being 'old' hold up given these estimates?

26. p21 (11): bracket is opened but not closed

27. p21 (14-16): repeated from before?

28. p21 (18): and we will move

29. p22 (6): remove comma

30. Study Design Table: I do not feel strongly about this, and but it seems to me that most of this table is superfluous because it largely repeats what is in the text (an exception is the explicit models that are fit in the 'Analysis Plan' column). I would urge the authors to consider whether this table is necessary.

---

## [Editor Report · Decision Letter 2]

3 Aug 2023

Dear Dr Tregenza,

Thank you for your patience while we considered your revised manuscript "Impacts of parental age and inbreeding on fitness in a wild insect" for publication as a Preregistered Research Article at PLOS Biology. This revised version of your manuscript has been evaluated by the PLOS Biology editors and by the Academic Editor.

Based on our Academic Editor's assessment of your revision, we are likely to accept this Stage 1 manuscript for publication-in-principle, provided you satisfactorily address the remaining points raised by the Academic Editor.

IMPORTANT: The Academic Editor still has some minor residual concerns about the clarity regarding details of the proposed design and analysis. Because of the complex formatting of his/her comments, I've provided them in an attached Word file rather than pasting them as flat text. Please attend to these concerns.

We expect to receive your revised manuscript within two weeks. 

*Published Peer Review History*

Sincerely,

Roli Roberts

Roland G Roberts, PhD

Senior Editor

PLOS Biology

rroberts@plos.org

on behalf of

Ines Alvarez-Garcia, PhD

Senior Editor,

ialvarez-garcia@plos.org,

PLOS Biology

---

## [Editor Report · Decision Letter 3]

2 Sep 2023

Dear Dr Tregenza,

Thank you for your patience while your revised Pre-Registered Research Article entitled "Impacts of parental age and inbreeding on fitness in a wild insect" was re-reviewed for PLOS Biology. Your Stage 1 manuscript has been evaluated by the PLOS Biology editors and by the Academic Editors.

We are happy to issue a Stage 1 'in-principle acceptance' decision, with a commitment to publish the final Stage 2 Preregistered Research Article (after revision, if needed), pending successful completion of the study. Please carefully read all the following information.

The study should now be completed according to the Stage 1 approved methods and analytic procedures, and the final manuscript should include an evidence-based interpretation of the results. Please see the review criteria for Stage 2 manuscripts here:

https://journals.plos.org/plosbiology/s/reviewer-guidelines#loc-reviewing-preregistered-research-articles

Subsequent editorial decisions for this study will not be based on the perceived importance or novelty of the results obtained during the data gathering and analysis phase of the work. It is critical however that you adhere to the approved Stage 1 study design when performing the study. Any deviation from these experimental procedures would need to be justified and approved by the editors (and potentially the reviewers), as otherwise it could lead to rejection of the manuscript at Stage 2. Please consult the editors immediately for advice if you need to alter this approved study plan.

**IMPORTANT**: Please follow the link below for important information regarding the Stage 2 manuscript template and review criteria. Please carefully read the guidelines on Stage 2 data collection BEFORE performing your study and completing your Stage 2 manuscript. 

AUTHOR GUIDELINES: https://genweb.plos.org/Marketing/Biology%20Preregistered%20Articles%20Guidelines%20for%20Authors.pdf

*Depositing this Stage 1 Protocol*

PLOS Biology does not publish Stage 1 Protocols immediately following an in-principle acceptance. Instead they are held and integrated into a single, completed 'Preregistered Research Article' following review and acceptance of the final Stage 2 manuscript. You are however required to register this approved Stage 1 Protocol with the Center for Open Science (https://cos.io/prereg/) or another recognised repository. This may be done publicly or under private embargo until submission of the Stage 2 manuscript. Stage 1 Protocols can be quickly and easily registered using a tailored mechanism for Registered Reports (https://osf.io/rr/). Please do this now. You will need to include the URL to this deposited protocol in your Stage 2 manuscript.

*Timeline*

We understand that carrying out the study will require a significant length of time and are willing to allow you 2 years to perform the study. Please email us at plosbiology@plos.org to discuss this if you have any questions or concerns, or to discuss an alternate timeline.

At this stage, your manuscript remains formally under active consideration at our journal. Please notify us by email if you do not wish to submit a Stage 2 manuscript or wish to pursue publication elsewhere, so that we may withdraw your manuscript. 

*Resubmission Checklist*

Before submitting the Stage 2 manuscript, please review the following resubmission checklist: https://plos.io/Biology_Checklist

Please note that for PRA stage 2, the response to reviewers file does not follow the standard format, but should rather be a document for the reviewers detailing the changes made to the manuscript since the stage 1 accept.

*Published Peer Review*

*PLOS Data Policy*

Please note that as a condition of publication, PLOS' data policy (http://journals.plos.org/plosbiology/s/data-availability) requires that you make available all data used to draw the conclusions arrived at in your manuscript. Please note that for this article type, the raw data itself should be archived and made freely available in a public repository rather than submitted as supplementary material. Please make sure to read the Stage 2 submission guidelines online regarding how this data should be annotated and appropriately time stamped to show that data was collected after this Stage 1 in-principle acceptance and not before.

*Blot and Gel Data Policy*

To enhance the reproducibility of your results, we recommend that, if applicable, you deposit your laboratory protocols in protocols.io, where a protocol can be assigned its own identifier (DOI) such that it can be cited independently in the future. For instructions see: https://journals.plos.org/plosbiology/s/submission-guidelines#loc-materials-and-methods

Thank you again for your submission to PLOS Biology. We hope that our editorial process has been constructive thus far, and we welcome your feedback at any time. Please don't hesitate to contact us if you have any questions or comments.

Sincerely,

Ines

--

Ines Alvarez-Garcia, PhD

Senior Editor

PLOS Biology

---

## [Decision Letter · Decision Letter 4]

8 Jan 2026

Dear Dr Tregenza,

Thank you for your patience while your manuscript entitled "Impacts of parental age and inbreeding on fitness in a wild insect" was peer-reviewed at PLOS Biology as a Preregistered Research Article. It has now been evaluated by the PLOS Biology editors, an Academic Editor with relevant expertise, and by the two original reviewers from the Stage I.

The reviews are attached below. As you will see, the reviewers praise the way the experiments have been conducted and the accuracy in following the pre-registered plan. However, they also raise several concerns regarding the structure and interpretation of the results. Reviewer 1 finds the introduction and methods too long and repetitive and that the conclusion that parental age at breeding has little or no effect on offspring performance is surprising, and could have an alternative explanation. In addition, the reviewer suggests a potential mechanism for the potential effect of hatching failure in the context of inbreeding. Reviewer 2 also raises several issues regarding the adherence to the pre-registration format, missing parental longevity covariate, inadequate justification that the age range includes senescent individuals, design limitation (confounded maternal/paternal age) not properly addressed, and weak conceptual development (no clear predictions despite available theory).

While we don’t disagree with the issues raised, the format previously agreed for the Preregistered Research Article to describe the experimental set up in the Introduction and Results must not be changed. However, additional analyses that deviate from the original plan can be undertaken as far as they are reported transparently as exploratory/post hoc and added in a new section. Please note that any additional analyses undertaken must be vital to support the current conclusions, and design limitations should be addressed in the Discussion section.

Given the extent of revision needed, we cannot make a decision about publication until we have seen the revised manuscript and your response to the reviewers' comments. Your revised manuscript is likely to be sent for further evaluation by all or a subset of the reviewers.

**IMPORTANT - SUBMITTING YOUR REVISION**

3. Resubmission Checklist

a) *PLOS Data Policy*

b) *Published Peer Review*

Sincerely,

Ines

--

Ines Alvarez-Garcia, PhD

Senior Editor

PLOS Biology

Reviewers' comments

Rev. 1:

This pre-registered study investigated effects of parental age, inbreeding, and ambient temperature on the fitness of female and male offspring in the wild. As the authors note, few studies have tested such effects in wild insects, which makes this study quite valuable. The results are somewhat difficult to interpret, which is not surprising for a field study. Nonetheless, I have some concerns about the analysis and interpretation. I also believe that the presentation could be improved.

Major comments:

The manuscript accurately describes the pre-registered plan. However, in terms of presentation, the Introduction and Methods sections are very long, wordy and quite repetitive in places, making the manuscript difficult to read and understand. The Introduction could perhaps be streamlined to avoid jumping back and forth between topics. It could also be shortened quite a bit. The Methods section repeats the same information in several places (e.g. L281-284). The long outline of the unsuccessful plan to measure offspring produced by the same females from eggs laid 20 days after the initial egg collection could be cut. Just explain briefly that within-individual comparison was not possible as a result of hatching failure of the late-life eggs.

I was surprised by the authors' conclusion that parental age at breeding had little or no effect on offspring performance. Almost complete failure of eggs from the late-life egg collection to hatch could be interpreted as a very strong, negative parental age effect on offspring viability. It appears that the effects of advanced parental age were highly deleterious for embryos, resulting in the death of nearly all late-life eggs. Many insects continue to lay eggs at advanced ages when almost none of the eggs are viable. Perhaps these crickets were doing the same. How else could this hatching failure be interpreted?

The authors do acknowledge and discuss the potential effect of hatching failure in the context of inbreeding, and acknowledge that much lower survival of inbred offspring could have biased the sample of adults. A potential mechanism for this that's perhaps worth mentioning more explicitly is that individuals homozygous for strongly deleterious recessive alleles would have died as embryos, resulting in an adult sample consisting of individuals that did not express such alleles. Another potential reason for the lack of inbreeding effects on adults could be that the source population had recently experienced a bottleneck. In such a case, many deleterious recessive alleles would likely have been purged in previous generations.

The authors housed juvenile crickets at high and low altitudes, but did not include altitude as a treatment in their analysis. This seems problematic because altitude appears to have affected both nymph survival and adult body size. I think altitude should be included in the models for the various fitness-related traits.

Other:

L220: Why "additionally"?

When there were not enough cameras for every active burrow, how did you choose which burrows would have cameras?

For Fig. 3 and Fig. 4, please explain what the points and whiskers mean. I guess this is mean +/- SE?

Rev. 2:

SUMMARY

This study represents a valuable contribution to understanding inbreeding and parental age effects in wild insects, a research area where field studies are exceptionally rare. The authors deserve credit for undertaking the substantial logistical challenge of conducting experimental manipulations in a natural population and for pre-registering their study.

However, the manuscript requires substantial revision. The primary issues are: (1) overly rigid adherence to pre-registration format that obscures what was accomplished, (2) missing parental longevity covariate (standard analytical approach), (3) inadequate justification that the age range includes senescent individuals, (4) design limitation (confounded maternal/paternal age) not properly addressed, and (5) weak conceptual development (no clear predictions despite available theory).

The manuscript needs restructuring to clearly present the study conducted (cross-sectional age variation plus experimental inbreeding) rather than remaining constrained by pre-registered components that failed (within-individual age comparison, temperature manipulation). With appropriate revisions, this could be an important contribution.

Recommendation: Major revisions required. Given the fundamental nature of some issues, I recommend substantial revision before detailed line-by-line review.

OVERARCHING ISSUE: Overly Rigid Application of Pre-Registration

The authors treat pre-registration as a constraint on clear presentation rather than a commitment to transparency. The temperature manipulation and within-individual age comparison produced no data (eggs didn't hatch), yet these are described extensively in Methods, illustrated in Figure 1 ("unchanged from original submission"), and discussed as if relevant. This obscures what was actually measured.

The Methods describe analyses as "the approach described in the first half of our design table" rather than simply describing what was done, making the section unnecessarily difficult to follow.

Pre-adult survival shows a 40% inbreeding effect (larger than any adult effect) but is relegated to another paper because it "was not part of our pre-registered plan." However, survival to adulthood is necessarily measured and well-known to be affected by inbreeding. This should have been anticipated.

In my view, pre-registration should commit to publishing results regardless of outcomes while allowing clear, complete presentation. It should not require extensive description of failed experiments, prohibit reporting important patterns in necessarily collected data, or force confusing organization.

Required: Restructure to clearly present the study conducted. Briefly acknowledge what failed (one Methods paragraph suffices). Report all major patterns including pre-adult survival. Organize for reader comprehension. Present Introduction and Discussion matching the realized design.

CRITICAL ISSUES

1. Missing parental longevity covariate

The authors maintained parents until death but did NOT include parental longevity as a covariate. This is standard practice for controlling selective disappearance and separating aging from quality effects, appropriate even in the absence of longitudinal data (van de Pol and Verhulst 2006; Ivimey-Cook and Moorad 2020, both cited). Even with only 10% mortality by age 39, quality heterogeneity among survivors (reflected in eventual longevity) could mask age effects. Parents who reproduced at age 30 and died at 40 days differ fundamentally from those who reproduced at 30 and died at 70 days.

Required: Re-analyze including parental longevity as a covariate (continuous or categorical). Report whether age effects remain null after controlling for parent quality.

2. No justification that age range includes senescent individuals

My Stage 1 Comment #19 asked for justification that the age range includes "old" individuals, suggesting survival curves or evidence of senescent declines at these ages. The authors cite "senescent declines" without specifying ages and state mean lifespan is 29 days. However, only 19/196 parents (10%) died by age 39. If 90% survive to age 39, calling 39-day-olds "old" is questionable. The age range may span young to middle-aged adults, not young to senescent.

This matters because null results could reflect true absence of effects OR insufficient age range that doesn't actually test senescent reproduction.

Required: Provide survival curves for this population. Specify ages at which senescent declines occur in cited studies. Explicitly justify that age range 6-39 days includes biologically "old" individuals. Reconcile 10% mortality by age 39 with mean lifespan of 29 days. If evidence cannot be provided, acknowledge this as a major limitation.

3. Incorrect statement about cross-sectional studies

The text states "previous cross-sectional studies may severely underestimate the effects of parental age on offspring quality," implying ALL such studies are inherently flawed. However, van de Pol and Verhulst (2006), which the authors cite, shows cross-sectional studies CAN properly estimate aging effects if individual longevity is included as a covariate. The authors then conduct a cross-sectional study without using this solution.

Required: Revise to specify that cross-sectional studies WITHOUT longevity covariates may underestimate effects. Acknowledge the current study is cross-sectional and apply the longevity covariate approach.

4. Contradiction about defining "old"

The Introduction argues for "experimental studies in which selective disappearance can be ruled out (i.e. there is no significant parental mortality until after the oldest parents have reproduced)." This is contradictory: if there's no significant mortality by when "oldest" parents reproduce, those individuals aren't experiencing senescence (characterized by elevated mortality). One cannot claim both that minimal mortality validates the design AND that the study examines senescent parents.

Required: Reconcile this contradiction. Provide survival curves showing when senescence begins. If the age range doesn't include senescent individuals, acknowledge you're not testing parental aging effects.

5. Figure 1 shows failed design

Figure 1 ("unchanged from original submission") shows temperature manipulation, second egg collection, and within-individual comparison. None produced data. Readers will misunderstand this as a within-individual experimental manipulation with temperature effects. The actual study is cross-sectional natural age variation plus inbreeding manipulation at a single time point.

Required: Revise Figure 1 to show the realized design or create two panels ("Planned" and "Realized") so readers understand what produced the data.

6. Confounded maternal and paternal age

Maternal and paternal age are correlated (tau = 0.53) because pairs were kept together. The authors assert this creates "high collinearity" justifying separate models. However, tau = 0.53 is moderate correlation, not extreme collinearity preventing estimation. Critically, the authors don't report attempting a full model with both ages. Incidentally, I don't think that tau is necessarily the most appropriate measure: Pearson's r tells you about the linear relationship between the variables, which is exactly what matters for collinearity in linear regression.

The separate models misleadingly suggest distinct, independently tested effects. Each still estimates a confounded effect. The design was avoidable (could have paired young males with old females and vice versa).

Required: Run a model including both maternal and paternal age. If it converges with stable estimates, report it (allows testing whether effects differ). If it fails, report this and acknowledge the design cannot distinguish maternal from paternal effects. Reframe results accordingly. Remove "unsurprisingly" when describing the correlation, which obscures that it results from design choice.

MAJOR ISSUES

7. No predictions despite available theory

The Discussion states "We did not have a strong prediction in relation to effects of parental age." This is problematic given: (1) evolutionary theory (Moorad and Nussey PNAS 2015 and others) predicts negative effects because selection weakens on traits expressed in offspring of older parents, (2) physiological mechanisms (mutation accumulation, oxidative damage, epigenetic changes) predict negative effects, (3) empirical data from G. bimaculatus (cited in Introduction) shows negative effects of parental age, (4) comparative data (Ivimey-Cook and Moorad 2020) shows 106 insect studies mostly finding negative effects, and (5) the authors made clear predictions for inbreeding based on similar logic.

Re: "Strong" predictions. Either you have a prediction or you don't. This suggests weak engagement with theory and makes the study appear exploratory rather than hypothesis-driven.

Required: State clear predictions based on theory and comparative data, then discuss null results in that context. Or explicitly justify why this species should differ from theoretical predictions and related species.

8. Interaction justification overstated

The text states the design "has allowed us to examine potential interactions." However, the pre-registration acknowledged interactions were "not a core aim" due to insufficient power. No interactions were found.

Required: Present this as two parallel investigations examined together for logistical efficiency, not as a study motivated by interaction testing.

9. Mismatch between motivation and measurement (Stage 1 Comment #12 inadequately addressed)

Stage 1 Comment #12 asked for clarity about what the study measures. The Introduction states "We are desperately short of information about how great the risk of inbreeding is in wild insects, and about its potential to depress population viability," invoking population-level concerns. However, the study does not measure inbreeding prevalence (how often it occurs) or population-level risk (prevalence times severity). It experimentally creates full-sib matings to measure severity of inbreeding depression when it occurs.

Required: Revise the Introduction to clearly frame the study as measuring the cost or severity of inbreeding depression when it occurs in this species, which is one component (but not the only component) of assessing population-level risk.

10. Overly complex explanations

The explanation of why annual species are useful remains convoluted despite Stage 1 Comment #22. The key point is simple: in species where generations don't overlap, fitness equals lifetime reproductive success. The current explanation involving "population growth rate parameters" obscures this. Similarly, the discussion of "alternative life-history strategies" versus "selective disappearance" is confusing because they appear to describe the same phenomenon (quality heterogeneity among individuals).

Required: Simplify these explanations.

MODERATE ISSUES

Methods organization: Study system description should precede experimental design. Readers need biological context before encountering protocols.

Evidence for senescence not used: The authors cite evidence for senescent declines but don't specify at what ages, missing an opportunity to justify their age range.

Paragraph breaks missing: New topics begin without breaks in several places.

"WildCrickets" undefined: This term appears without explanation.

Terminology: "Standard normal standardization" should be "standardized to mean 0 and SD 1" with brief explanation of why.

Distribution families not justified: Authors assert choices are "most appropriate" without explaining why (e.g., why negative binomial over Poisson for counts? Was overdispersion tested?).

Pre-adult survival deferred: The 40% survival reduction is larger than any adult effect but is relegated to another paper. While detailed age-specific analysis might warrant separate treatment, basic survival-to-adulthood statistics should be reported here as this is a fundamental fitness component necessarily measured.

STAGE 1 OVERSIGHTS

Several issues should have been caught at Stage 1:

Confounded maternal/paternal age: I failed to identify this design flaw. The design could have created independent variation by pairing young males with old females and vice versa. This cannot be fully fixed now but can potentially be salvaged by running the full model including both ages. If that fails, the limitation must be acknowledged.

Weak dual-factor justification: The conceptual motivation for studying both inbreeding and parental age together needed clearer development, especially given that interaction testing was not a core aim.

Pre-adult survival not pre-registered: Survival to adulthood is necessarily measured and well-known to be affected by inbreeding. This fundamental fitness component should have been included in pre-registered outcomes.

CONCLUSION

This represents valuable work in a challenging system where field studies are rare. With substantial revision addressing the critical issues (particularly adding parental longevity covariate, justifying age range, attempting full model for maternal/paternal age, engaging with theory, and reporting pre-adult survival), this could be an important contribution to understanding inbreeding and parental age effects in wild insects. The null results for parental age would be particularly interesting if properly justified and interpreted.

I am willing to provide detailed line-by-line review of a revised manuscript once these fundamental issues are addressed.

---

## [Editor Report · Decision Letter 5]

20 Feb 2026

Dear Dr Tregenza,

Thank you for your patience while we considered your revised manuscript entitled "Impacts of parental age and inbreeding on fitness in a wild insect" for publication as a Preregistered Research Article at PLOS Biology. This revised version of your manuscript has been evaluated by the PLOS Biology editors and by the two Academic Editors.

Based on our Academic Editors' assessment of your revision, we are likely to accept this manuscript for publication, provided you satisfactorily address the data and other policy-related requests stated below my signature. In addition, you should address the following points:

1) With regards to Reviewer 2's suggestion that male age and female age should be treated as separate predictors rather than using mean parental age, one of the Academid Editor has suggested adding a single sentence to the Discussion in the Age section saying:

'Treating male and female age as seperate predictors rather than using mean age did not change our conclusion that parental age does not affect any of the measured traits (Data is provided for any reader wishing to confirm this conclusion)."

2) Please also add a footnote in Fig. 1 legend explaining that the area marked with crosses indicates a component of the planned study that had to be modified.

We expect to receive your revised manuscript within two weeks. 

*Published Peer Review History*

*Press*

Sincerely,

Ines

--

Ines Alvarez-Garcia, PhD

Senior Editor

PLOS Biology

DATA POLICY:

Fig. 2; Fig. 3 and Fig. 4

---

## [Editor Report · Decision Letter 6]

2 Mar 2026

Dear Dr Tregenza,

Thank you for the submission of your revised Preregistered Research Article entitled "Impacts of parental age and inbreeding on fitness in a wild insect" for publication in PLOS Biology. On behalf of my colleagues and the Academic Editors, Michael Jennions and Chris Chambers, I am delighted to let you know that we can in principle accept your manuscript for publication, provided you address any remaining formatting and reporting issues. These will be detailed in an email you should receive within 2-3 business days from our colleagues in the journal operations team; no action is required from you until then. Please note that we will not be able to formally accept your manuscript and schedule it for publication until you have completed any requested changes.

PRESS

Sincerely,

Ines

--

Ines Alvarez-Garcia, PhD

Senior Editor

PLOS Biology
